# A Novel Role for the Soluble Isoform of CTLA-4 in Normal, Dysplastic and Neoplastic Oral and Oropharyngeal Epithelia

**DOI:** 10.3390/cancers15061696

**Published:** 2023-03-10

**Authors:** Prarthna Clare, Farah Al-Fatyan, Badri Risheh, Kristine Nellany, Frank James Ward, Rasha Abu-Eid

**Affiliations:** 1Institute of Dentistry, School of Medicine, Medical Sciences & Nutrition, University of Aberdeen, Aberdeen AB25 2ZR, UK; 2Institute of Medical Sciences, School of Medicine, Medical Sciences & Nutrition, University of Aberdeen, Aberdeen AB25 2ZR, UK; 3NHS Grampian Biorepository, Aberdeen Royal Infirmary, Aberdeen AB25 2ZN, UK; 4Aberdeen Cancer Centre, University of Aberdeen, Aberdeen AB25 2ZR, UK

**Keywords:** sCTLA-4, CTLA-4, head and neck cancer, oral epithelial dysplasia, oral potentially malignant disorders, immune checkpoints, image analysis

## Abstract

**Simple Summary:**

Head and neck cancers have a poor prognosis mainly attributed to late diagnosis when a cure is not possible. Markers that are capable of predicting cancerous changes at an early stage are needed. Here, we quantified the soluble isoform of CTLA-4 (sCTLA-4), a molecule usually described for its immune function, in normal, potentially malignant and malignant oral and oropharyngeal epithelial samples. We report a distinct sCTLA-4 staning pattern and distribution in normal samples indicative of a new role for this molecule in epithelial cell growth and development. We also describe significant changes in sCTLA-4 staining in potentially malignant and malignant samples, suggestiong the potential of sCTLA-4 as a predictor of disease progression.

**Abstract:**

*Background*: Head and neck cancer (HNC) has a high mortality rate, with late diagnosis remaining the most important factor affecting patient survival. Therefore, it is imperative to identify markers that aid in early detection and prediction of disease progression. HNCs evade the immune system by different mechanisms, including immune checkpoints. Cytotoxic T-lymphocyte-associated protein-4 (CTLA-4) is an immune checkpoint receptor that downregulates anti-tumour immune responses, with evidence of involvement in HNC. The less studied, alternatively spliced, soluble isoform (sCTLA-4) also plays an immunosuppressive role that contributes to immune escape. We quantified sCTLA-4 in normal, potentially malignant, and malignant oral and oropharyngeal tissues to elucidate any role in tumourigenesis and identify its potential as a biomarker for diagnosis and patient stratification. *Methods:* Normal, low- and high-grade epithelial dysplasia, and squamous cell carcinoma oral and oropharyngeal biopsies were selectively stained for sCTLA-4 and quantified using the image analysis software QuPath. *Results:* Distinct sCTLA-4 staining patterns were observed, in which normal epithelial sCTLA-4 expression correlated with keratinocyte differentiation, while disrupted expression, both in intensity and localisation, was observed in dysplastic and neoplastic tissues. *Conclusions:* Our data indicate an additional, previously unknown role for sCTLA-4 in epithelial cell differentiation and proliferation. Furthermore, our findings suggest the potential of sCTLA-4 as a biomarker for predicting disease progression and patient stratification for targeted HNC therapies.

## 1. Introduction

Head and neck cancer (HNC) encompasses malignancies of the oral cavity, oropharynx, hypopharynx, larynx, and salivary glands, with 90% being squamous cell carcinoma (SCC). In 2020, the estimated global number of new cases was 377,713 for lip and oral cavity cancer and 98,412 for oropharyngeal cancer, with deaths estimated at 177,757 for the former and 48,143 for the latter [1].

The incidence of HNC is rising globally, with variation between countries arising from different risk factors, including smoking, excessive alcohol consumption, smokeless tobacco, and betel nut chewing [1]. A subtype of HNC for which the primary risk factor is the human papilloma virus (HPV) has recently increased in incidence [2,3,4] and is reported in a younger age group.

Despite treatment advancements, HNC, particularly oral cancer, has seen little improvement in survival rates. This is mainly attributed to late diagnosis, with one of the main limiting factors being the inability to reliably predict malignant transformation in oral potentially malignant disorders (OPMDs), which carry an increased risk of transforming into malignancy [5].

Histologically, oral cancers are commonly preceded by a range of cellular alterations, termed oral epithelial dysplasia (OED). OED grading systems present a wide variation in their predictive value, mainly because of the subjective evaluation of parameters used to define dysplasia, a lack of calibration, and an absence of validated criteria important for predicting malignant transformation [6]. Therefore, there is an urgent need to identify biologically meaningful objective markers capable of predicting malignant changes.

The tumour microenvironment in HNC is intrinsically immunosuppressive and is influenced by suppressive immune cells, regulatory checkpoints, and cytokines. Suppressive immune checkpoints such as programmed death-1 (PD-1) and cytotoxic T-lymphocyte associated antigen-4 (CTLA-4) can be exploited either directly or indirectly by tumour cells to avoid elimination by the anti-tumour immune response. In particular, the inhibitory receptor CTLA-4 is critical for maintaining immune homeostasis, as highlighted by CTLA-4^KO^ mice, which die soon after birth from expanded effector T lymphocyte infiltrates in various organs [7,8]. Further evidence of its relevance comes from individuals with autosomal dominant dysregulation of CTLA-4 function associated with immune-mediated pathology [9].

In cancers, CTLA-4 extrinsically controls effector T-cell responses, a mechanism mediated by regulatory CD4 T-cells (Tregs), and in the context of cancer strategies to evade anti-tumour immunity, there is evidence that tumours can actively induce CTLA-4^+^ intratumoural Tregs to propagate an immunosuppressive milieu, thereby preventing effector T-cells from initiating successful anti-tumour responses. This mechanism is to some extent facilitated by the less well studied, alternatively spliced secretable isoform, soluble CTLA-4 (sCTLA-4) [10,11], which, like CTLA-4, has a direct suppressive effect on T-cell activation by binding to the B7 ligands on APC [12]. Serum levels of sCTLA-4 are high in several cancers, but it has only recently been identified to have an immunosuppressive capacity like recombinant CTLA4-Ig, an inhibitory costimulatory modulator used to treat rheumatoid arthritis [13,14]. In some cancers, there is evidence that tumour cells express high levels of sCTLA-4, probably to prevent effector T-cell activation directly, providing a novel mechanism for tumour escape. In a recent systematic review, we provided strong evidence that malignant cells express variable levels of CTLA-4 (transcripts and functional) on the cell surface or within the cytoplasm in many types of cancer, including HNC (laryngeal and pharyngeal) [15].

CTLA-4 is an established target for immunotherapy [16]. However, antibodies that target the CTLA-4 receptor also bind the sCTLA-4 isoform. The potential for selectively targeting sCTLA-4 has not been explored, and its role is not fully understood, although we have previously demonstrated that selective blockade of sCTLA-4 does induce anti-tumour activity in some murine models [12].

In this study, we used sCTLA-4-specific antibodies, which cannot bind the CTLA-4 receptor isoform, to selectively investigate sCTLA-4 expression in normal, potentially malignant, and malignant oral and oropharyngeal tissues. Our data suggest that sCTLA-4 is expressed in both healthy and diseased non-immune epithelial cells, and that quantification of sCTLA-4 as a biomarker has the potential to detect early disease progression and serve as a means for patient stratification.

## 2. Materials and Methods

### 2.1. Immunofluorescence

#### 2.1.1. Cell-Lines

Human HNC-SCC cell lines were purchased from the American Tissue Culture Collection (ATCC). FaDu (ATCC^®^ HTB-43™) was derived from the pharynx, and both CAL 27 (ATCC^®^ CRL-2095™) and UPCI: SCC154 (which is HPV positive) (ATCC^®^ CRL-3241™) were from the tongue. Cells were incubated at 37 °C and 5% CO_2_. Eagle’s Minimum Essential Medium (BD, USA) was used to culture FaDu and SCC154, and Dulbecco’s Modified Eagle’s Medium (Lonza, Slough, UK/Corning, Manassas, VA, USA) was used for CAL27. The media were supplemented with foetal bovine serum (10%) (Biosera, Cholet, France), L-glutamine (2 mM), penicillin (100 U/mL), and streptomycin (100 μg/mL) (Gibco, Waltham, MA, USA). The culture medium was replaced three times a week, and cells were checked regularly.

#### 2.1.2. Anti-sCTLA-4 Antibodies

Two anti-sCTLA-4 antibodies were used in this study. The first antibody used for confocal microscopy, a biotinylated selective 73-B1 anti-sCTLA-4 IgG1κ mAb, was a kind gift from Dr David Matthews (MRCT). Mab 73-B1 specificity was confirmed by sandwich ELISA using recombinant human sCTLA-4 (EC_50_:24 nM) and lack of specificity for the full-length receptor isoform by flow cytometric analysis of full-length CTLA-4-transfected HEK293 cells (unpublished data). To ensure the sensitivity of the antibody, the CTLA-4 gene encoding both isoforms was knocked down using siRNA (10782097 (4392420)) in A549 lung adenocarcinoma cells (Appendix A).

For immunohistochemistry staining, JMW-3B3, a selective IgG1λ anti-human sCTLA-4 monoclonal antibody developed in house, was used. JMW-3B3 has been fully characterized, and its specificity to sCTLA-4 has been reported [12].

The two antibodies are interchangeable, as both have been raised against the same C-terminal sCTLA-4-specific antigen, with the difference being that 73-B1 has a higher affinity than JMW-3B3 for human sCTLA-4.

#### 2.1.3. Staining

Cells were detached and seeded in Ibidi microslide 8-well plates (Thistle Scientific, Glasgow, UK) and incubated at least overnight (37 °C, 5% CO_2_) until the desired confluency was reached. Cells were then washed twice with PBS, fixed with Cytofix (BD Biosciences, Oxford, UK) (10 min, RT), and 0.3 M glycine (Fisher Scientific, Loughborough, UK) in PBS was added (5 min, RT) to prevent background fluorescence. This was followed by permeabilization with 0.2% Triton X-100 in PBS (Sigma Aldrich, UK) (5 min, RT) and blocking with 1% BSA (Fisher Bioreagents, Pittsburgh, PA, USA) in PBS (10 min, RT). Cells were washed with PBS 2–3 times (2 min) after each step. Cells were then incubated with 20 μg/mL of the 73-B1 anti-sCTLA-4 IgG1κ mAb or biotinylated mouse IgG1κ isotype control (Invitrogen, Warrington, UK) and incubated overnight at 4 °C. Cells were washed twice with PBS, and streptavidin-AF555 was added (Invitrogen, UK) to selectively reveal sCTLA-4, according to the manufacturer’s instructions (30 min, RT in the dark). After three washes, 150 µL of diluted DAPI (Invitrogen, UK) in PBS (1:1000 of 0.1 µg/mL) was added and incubated for 5 min, followed by three washes.

#### 2.1.4. Imaging

Cell imaging was performed with a LSM880 confocal microscope with an oil immersion objective (X63) at the Institute of Medical Sciences, Microscopy and Histology Core Facility. Images from merged and separate channels were obtained using Zen blue (Zeiss, Jena, Germany) and analysed using ImageJ (v1.47v) [17] to quantify the sCTLA-4 intensity in the cytoplasm and nucleus. Details of the analysis are shown in Appendix A.

### 2.2. Immunohistochemistry

#### 2.2.1. Samples

Histological sections from the oral cavity and the oropharynx were obtained from the NHS Grampian Biorepository. Ethical approval was granted by the Biorepository Scientific Access Group (tissue requests TR000132 and TR000189), under IRAS project: 296502. The samples from the oral cavity were taken from the floor of the mouth and included normal (n = 7), low-grade dysplasia (LGD) (n = 16), high-grade dysplasia (HGD) (n = 9) and squamous cell carcinoma (SCC) (n = 5). Samples from the oropharynx included normal oral mucosa from the base of the tongue (n = 3), LGD from the tonsil (n = 1), HGD from the base of the tongue (n = 3), HPV-negative (HPV^−ve^) SCC from the base of the tongue (n = 3), and HPV-positive (HPV^+ve^) SCC from the base of the tongue (n = 3). The diagnosis of the samples was confirmed by a pathologist from the NHS Grampian Biorepository using diagnostic haematoxylin and eosin-stained sections.

#### 2.2.2. Immunohistochemistry Staining

Sections were dewaxed and dehydrated in xylene, ethanol, and industrial denatured alcohol. Antigen retrieval was achieved by heat-induced epitope retrieval through immersion in an ethylenediaminetetraacetic acid (EDTA) buffer (pH = 7.8) and heating in the microwave (10 min) twice. Tonsil sections were used as staining controls.

Staining was performed using a DAKO autostainer (Dako, Glostrup, Denmark), using the selective anti-sCTLA-4 primary antibody (JMW-3B3 at 20 μg/mL). The primary antibody was incubated (1 h, RT). A peroxidase enzyme block was performed, and immunostaining was visualised using 3,3′-diaminobenzidine (DAB) chromogen substrate. Slides were then immersed in a 0.5% copper sulphate solution, counterstained using Harris’ haematoxylin, further immersed in dilute lithium carbonate to enhance the counterstain, dehydrated, rinsed with xylene, and mounted.

#### 2.2.3. Imaging

Slides were scanned using a Zeiss Axio Scan.Z1 slide scanner (Carl Zeiss Microscopy, France) at the University of Aberdeen Microscopy and Histology Core Facility. Slides were scanned at X20 magnification (resolution: 0.22 μm/pixel). Whole slide scans (CZI format) were viewed using QuPath (v.0.1.2) [18]. Some sample slides included multiple tissue sections from the same patient, and each was analysed separately. This resulted in the total number of section images being: from the oral cavity, normal n = 10, LGD n = 28, HGD n = 13, and SCC n = 12; and from the oropharynx: normal n = 4, LGD n = 5, HGD n = 8, HPV^−ve^ SCC n = 25, and HPV^+ve^ SCC n = 8.

#### 2.2.4. Image Analysis

sCTLA-4 staining intensity in the epithelium was quantified using Qupath (v0.1.2) [18]. The epithelial compartment was annotated using semi-automated annotation tools. A watershed transform for cell detection based on the haematoxylin optical density (OD) (as haematoxylin stains the nuclei) was applied to the annotated epithelium.

Following cell detection, the staining intensity for sCTLA-4 was colour coded according to the mean cellular DAB OD. The gradient range was automatically produced according to the detected staining intensity values, and a detection classifier was created and “trained” by setting classes through sub-annotations classified as strong, intermediate, and weak staining. This enabled the classifier to automatically classify all the cells within the epithelium into classes based on their staining intensity.

In addition to the mean cell DAB OD used to colour code the staining intensity, the mean nuclear and cytoplasmic DAB OD values of positively stained cells were compared in different diagnostic groups to identify differences in sCTLA-4 between the two cell compartments.

Staining intensity percentages were compiled to create the H-score (histology score), a semi-quantitative scoring system used to evaluate immunohistochemistry staining. H-Score was calculated using the formula [19]: H-score = 3 × the percentage of strong staining + 2 × the percentage of moderate staining + the percentage of weak staining.

### 2.3. Statistical Analysis

Statistical analyses were performed using GraphPad Prism. Depending on data normality, non-parametric tests (Kruskal-Wallis with *post hoc* Dunn’s tests) or one-way analysis of variance (ANOVA) with a *post hoc* Tukey test were used to detect differences between groups. Depending on data normality, Mann-Whitney or unpaired *t*-tests were used to compare HPV^+ve^ and HPV^−ve^ SCC and to compare nuclear to cytoplasmic staining in tissue sections. *p* < 0.05 was considered statistically significant.

## 3. Results

### 3.1. Head and Neck Cancer Cell Lines Express sCTLA-4

To examine whether HNC cells expressed sCTLA-4, FaDU, CAL27, and SCC-154 cell lines were stained with anti-sCTLA-4 mAb (73-B1) and staining intensity was quantified using ImageJ. Appendix A shows the presence of sCTLA-4 in the cytoplasm and nuclei of all three cell lines, with intensified staining observed in the nuclei. Although not statistically significant, cytoplasmic sCTLA-4 expression appeared to be stronger in the HPV^+ve^ SCC154 cell line compared to HPV^−ve^ cell lines (Appendix A).

Quantification of sCTLA-4 staining intensity in nuclei showed a significantly higher level in FADU compared to CAL27 and SCC-154 (Appendix A). Isotype controls are shown in Appendix A.

### 3.2. sCTLA-4 Distribution and Intensity in Oral/Oropharyngeal Epithelia Change with Disease Progression

Given that HNC cell lines express sCTLA-4, we assessed patterns of expression in tissue biopsies from healthy, dysplastic and confirmed neoplastic specimens. A comparison of the differences in sCTLA-4 staining intensity in the epithelial compartment was performed using QuPath to generate a colour-coded representation of different staining intensities (red = strong staining, yellow = intermediate staining, and green = weak staining).

#### 3.2.1. Oral Tissues

When assessing samples from the oral cavity, a distinct staining pattern was observed within the normal epithelium. The staining intensity was the highest in the basal and parabasal layers, with intensity decreasing from high to moderate in the intermediate layers, and eventually little or no staining in the superficial layers of the epithelium (Figure 1 and Appendix A). Cases of dysplasia did not show a specific pattern, while SCC samples displayed more widely spread intermediate and strong staining (Figure 1, Appendix A).

While quantifying different staining intensities for particular diagnostic groups, normal and SCC samples had the highest percentage of strong staining in comparison to other staining intensities, while LGD and HGD exhibited the highest percentage of intermediate intensity whether analysed separately (Figure 2) or in combination (Appendix A).

When comparing the mean DAB OD between the nucleus and cytoplasm for all the diagnostic groups and all the staining intensities (not shown), the nucleus stained more strongly than the cytoplasm, similar to what was observed in HNC cell lines (Figure 2 and Appendix A). The biggest differences between nuclear and cytoplasmic staining were observed in normal and SCC samples (Figure 2), as well as in cases of dysplasia when LGD and HGD were combined as a single group (Appendix A).

Comparing specific staining intensities between different diagnostic groups showed that LGD had the lowest percentage of negative cells. Intermediate staining was the highest in dysplasia samples, while both normal and SCC samples displayed mainly strong staining (Figure 2, Appendix A). Although the percentage of strong staining was not different between normal and SCC samples, as described above, the distribution of staining was different. Normal basal cells and malignant cells are highly proliferating cells, indicating sCTLA-4’s potential involvement in cell proliferation (Figure 1 and Figure 2, and Appendix A).

Intermediate, followed by strong staining levels, showed the biggest differences between diagnostic groups (Figure 2, Appendix A). The H-score was the highest in normal and SCC samples (Appendix A), reflecting the observed staining patterns.

#### 3.2.2. Oropharyngeal Tissues

A similar pattern was observed in normal oropharyngeal tissues, with the basal and parabasal layers expressing the strongest staining intensity. However, this was not observed in all samples and was not as clearly demarcated as the tissue biopsies from the oral cavity. A striking feature was the predominance of intermediate staining intensity in dysplasia and in HPV^+ve^ samples, while the strongest staining was observed in HPV^−ve^ and normal samples (Figure 3 and Figure 4, and Appendix A). For each diagnostic group, nuclear staining was stronger than cytoplasmic staining, with the most significant differences observed in HGD and both HPV^−ve^ and HPV^+ve^ SCC (Figure 4).

A distinct staining pattern was noted in HPV^+ve^ cancer samples where there appeared to be small clusters, or “hot spots”, of strong staining amidst moderate staining intensity in the epithelium (Figure 3, Appendix A). This pattern was also observed in some normal (Appendix A) and dysplastic samples, suggesting they are potentially infected with HPV. However, since these tissues are not routinely tested for HPV, it is not possible to confirm this.

As HPV^−ve^ and HPV^+ve^ SCC vary significantly in their clinical behaviour, we compared sCTLA-4 staining in these two entities independent of normal and dysplastic samples. Interestingly, although the nuclear and cytoplasmic ODs of sCTLA-4 staining were significantly higher in HPV^+ve^ samples, they had a significantly higher percentage of weak and intermediate staining than HPV^−ve^ and a significantly lower H-index (Table 1), reflecting more diffuse strong staining in HPV^−ve^ samples.

### 3.3. Overall Findings

Our results showed that different HNC cell lines express sCTLA-4 in the cytoplasm and more strongly in the nucleus.

This finding led us to assess sCTLA-4’s expression in clinical tissues, and while we expected to see some level of sCTLA-4 in malignant and dysplastic tissues, unexpectedly, our results showed that sCTLA-4 is expressed in normal oral and oropharyngeal epithelial cells. The pattern of sCTLA-4 expression in normal oral epithelium follows the normal stratification of differentiating keratinocytes across different cell layers. This suggests sCTLA-4’s involvement in keratinocyte differentiation. A similar staining intensity in the proliferating layers of normal epithelium (basal and parabasal) and the highly proliferating malignant cells further suggests a role for sCTLA-4 in epithelial cell proliferation.

These patterns were not as clear in most normal oropharyngeal tissues, which could relate to the proximity to lymphoid tissues in an anatomical region subject to inflammation and therefore characterised by the expression of high levels of immune markers. As can be seen in Figure 5, normal oropharyngeal tissues with heavier inflammatory infiltrates directly under the epithelium tended to display widespread strong staining, while the normal sample in Figure 3, which had very few inflammatory cells, displayed a pattern that followed keratinocyte differentiation. A clear difference was observed between HPV^−ve^ and HPV^+ve^ SCC, with the former expressing more diffuse, strong staining and the latter displaying a more focal, strong expression and an overall lower staining intensity. This is consistent with the differences in the behaviour of these two distinct diseases, further suggesting a potential for sCTLA-4 to predict disease severity.

## 4. Discussion

HNC patients still suffer from high mortality rates, despite improvements in treatments. This is mainly due to late diagnosis at a stage when a cure is no longer possible. Identifying biomarkers that aid in early diagnosis and detection of malignant changes in OPMDs is a pressing need, especially with the global increase in HNC incidence.

The tumour microenvironment in HNC is immunosuppressive, which aids in tumour escape from immune responses. One such immune evasion mechanism is the checkpoint receptor CTLA-4. In addition to T-cells, CTLA-4 is constitutively expressed on solid human tumour cells such as certain carcinomas [20], and a wide range of cancers express variable levels of CTLA-4 [15].

There is evidence of a role for CTLA-4 in HNC; genetic polymorphisms of CTLA-4 were linked to susceptibility and prognosis for these malignancies [21,22,23], and CTLA-4 is involved in the suppressive function of Tregs [24], with Tregs expressing high levels of CTLA-4 in HNC being highly suppressive [25].

The less-known alternately spliced soluble isoform sCTLA-4 [26,27] could also contribute to immune evasion in cancers. Therefore, we quantified, for the first time, sCTLA-4 expression in HNC cell lines and in normal, potentially malignant, and malignant oral and oropharyngeal tissues.

Our results showed that sCTLA-4 expression varies based on the site (oral/oropharyngeal), HPV status, stage of the disease (normal/premalignant/malignant), and localisation (nuclear/cytoplasmic).

All three HNC cell lines that we assessed expressed cytoplasmic and nuclear sCTLA-4, with stronger nuclear staining. Stronger nuclear staining was also observed in all the epithelia, regardless of the diagnostic group and staining intensity.

In terms of tissue location, although normal oral and oropharyngeal epithelia displayed a characteristic pattern that correlates with keratinocyte differentiation, the distribution of strong sCTLA-4 expression was more widely spread in normal oropharyngeal tissues in comparison to oral tissues. Both tissues are constantly exposed to external stimuli, and with the abundance of lymphoid tissues in the oropharynx, accessing healthy oropharyngeal tissues that are completely devoid of inflammation is challenging. This could contribute to the absence of a clear sCTLA-4 staining pattern in many normal oropharyngeal epithelia as immune cells express high levels of sCTLA-4 [12] and could contribute to the diffuse high levels observed.

As for the diagnostic group, normal oral tissues showed a clear pattern of sCTLA-4 expression, with the strongest staining observed in the basal and parabasal layers, with intensity gradually decreasing corresponding to differentiation of the stratified epithelial layers. This contrasted with the widespread strong staining in SCC, while dysplastic lesions displayed the highest level of intermediate staining but showed no clear pattern.

The strongest staining intensity was observed in the basal and parabasal layers of the normal oral epithelium and throughout the malignant epithelium. The most pronounced differences between nuclear and cytoplasmic staining were also observed in these two groups. Both these tissues have high proliferative abilities, indicating a surprising potential role for sCTLA-4 in regulating healthy keratinocyte proliferation. The distinct distribution pattern of sCTLA-4 observed in normal epithelium in different cell layers suggests a further role for sCTLA-4 in keratinocyte differentiation, with the least differentiated cell layers (basal and parabasal) showing the highest level of expression. Interestingly, strong staining was also observed in a significant percentage of malignant epithelial cells that are characterised by a loss of control of normal keratinocyte differentiation. So far, sCTLA-4 has only been studied as an immune modulator, and our data infer a broader role for this molecule in epithelial cell differentiation and proliferation.

Markers for predicting disease progression in OED are still elusive. Here we showed that intermediate levels of sCTLA-4 staining intensity were higher in OED in contrast to stronger staining in normal and SCC tissues (with different distributions). This quantified change in staining intensity could differentiate dysplastic lesions from normal and SCC tissues. This shows a great translational potential for sCTLA-4 as a predictor of disease progression in OED, thus aiding in the early diagnosis of malignant changes.

When assessing the H-index as an indicator of overall staining intensity, in oral cavity samples, there was an increase in sCTLA-4 expression in SCC, especially in comparison to cases of dysplasia. This was also observed in oropharyngeal tissues, but only in HPV^−ve^ SCC, while HPV^+ve^ samples had lower levels of sCTLA-4.

This difference between oropharyngeal samples based on the HPV status was apparent when comparing different staining intensity parameters. These differences are consistent with the differences between these two entities in prognosis, response to therapy, and survival, where HPV^+ve^ cancers have a better clinical outcome [28,29].

A distinct sCTLA-4 staining pattern was observed in HPV^+ve^ samples, with “hot spots” of high intensity staining spread through the epithelium in the middle of the intermediate staining. We speculate that these hot spots could be individually infected cells (with HPV) or immune cells that infiltrate the epithelium and form clusters of cells expressing high levels of sCTLA-4. The latter are possibly Foxp3^+^ Tregs within the epithelium, which have been reported at higher levels in HPV^+ve^ tumours in comparison to HPV^−ve^ [30]. This speculation is supported by the fact that CTLA-4 is associated with the recruitment of Tregs into the tumour microenvironment [31] and that Tregs express high levels of CTLA-4 [32]. Some normal and dysplastic oropharyngeal lesions also showed this pattern, suggesting they could be HPV positive, however, HPV status is not routinely tested in non-neoplastic oropharyngeal samples. Observing these hot spots could be an indication of the infection and therefore useful in identifying patients at risk of developing future disease.

Here, we found that higher levels of sCTLA-4 expression were observed in the diagnostic groups with a worse prognosis in both oral (SCC) and HPV^−ve^ oropharyngeal tissues. We have previously reported that serum sCTLA-4 levels are elevated in melanoma patients compared with healthy volunteers [13], and given the differences we observed between different diagnostic groups, it would be of interest to assess the serum level of sCTLA-4 in oral and oropharyngeal disease cohorts as a potential immunosuppressant and correlate it to the tissue expression patterns.

Our findings show great clinical translational potential for sCTLA-4 in HNC and highlight the need for larger studies in OPMD and HNC to assess sCTLA-4 and its correlation to clinical outcome.

The distinct staining patterns and differences in expression levels of sCTLA-4 across the spectrum of normal, dysplastic, and neoplastic oral and oropharyngeal tissues suggest its potential as a marker to aid in the objective diagnosis and classification of these lesions. Advances in machine learning and artificial intelligence can further quantify this marker to present a biologically meaningful and clinically applicable biomarker that can be included in the diagnostic criteria for OED and SCC, as our data indicate a role in cell differentiation and proliferation.

In conclusion, we report differences in the expression of sCTLA-4 in normal, dysplastic, and neoplastic oral and oropharyngeal epithelia, indicating a role of sCTLA-4 in normal differentiation and proliferation of epithelial cells with disruptions associated with premalignant and malignant changes. This makes sCTLA-4 a potential marker for predicting disease progression. The differences in intensity and distribution between different diagnostic groups and indeed between HPV^+ve^ and HPV^−ve^ SCC, which have different clinical outcomes, suggest the potential of sCTLA-4 as a therapeutic target and as a marker for patient diagnosis, stratification, and treatment.

## Figures and Tables

**Figure 1 cancers-15-01696-f001:**
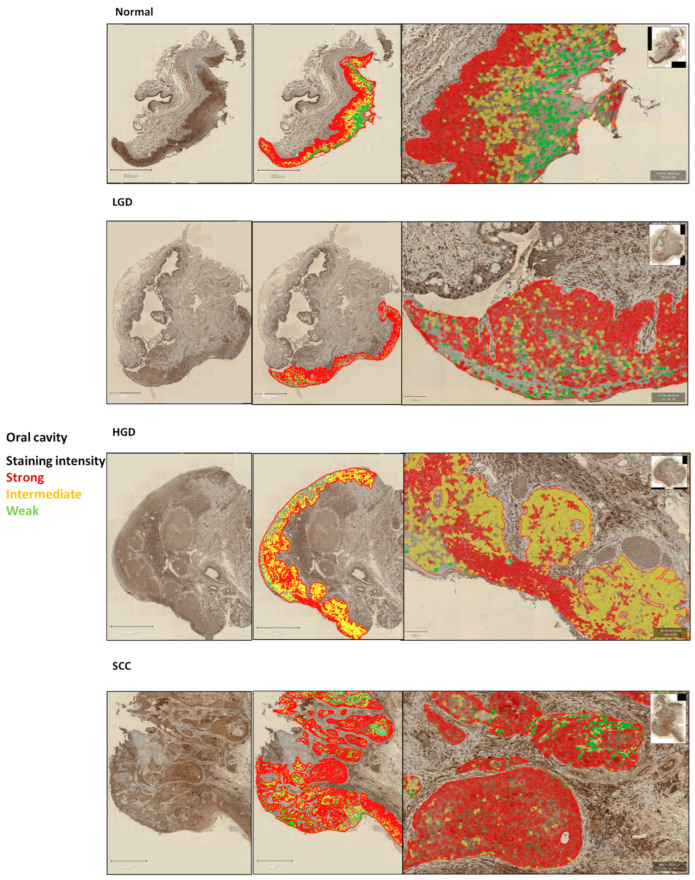
Representative sections from whole-slide scans (X20) of normal, low-grade dysplasia (LGD), high-grade dysplasia (HGD), and squamous cell carcinoma (SCC) from the floor of the mouth. The left panel shows the immunohistochemical staining for sCTLA-4 (DAB, brown) counterstained with haematoxylin (blue); the scale bar represents 800 µm. The middle panel shows the annotations and cell detections with colour-coded staining intensities for the same sections. The right panel represents zoomed-in views of the cell detection, with the scale bar representing 100 µm and the inset showing the exact site of the magnified view.

**Figure 2 cancers-15-01696-f002:**
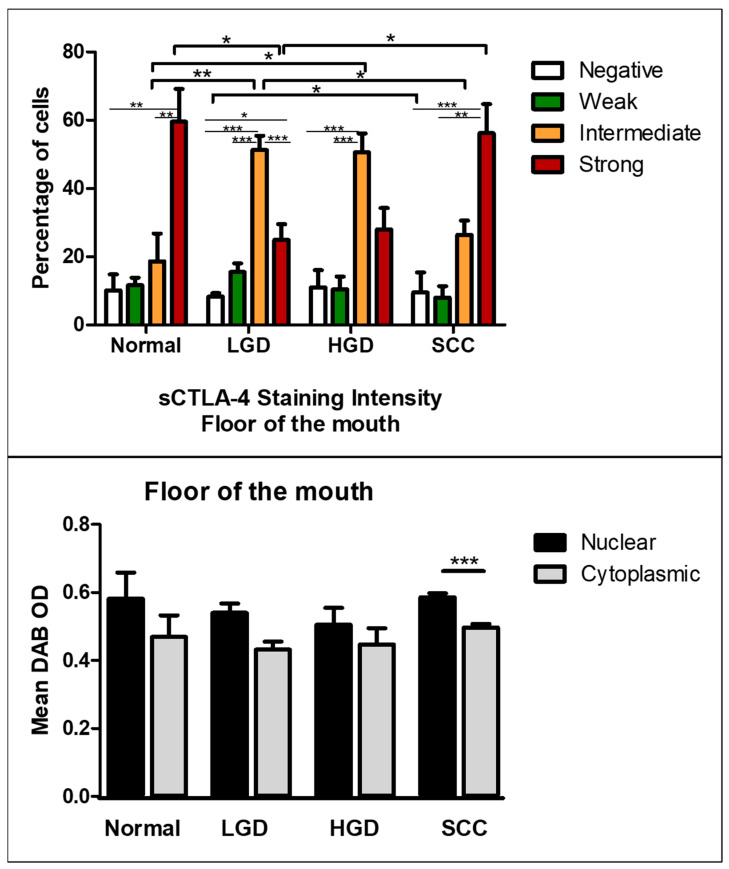
Quantification of the staining intensity in normal, low-grade dysplasia (LGD), high-grade dysplasia (HGD), and squamous cell carcinoma (SCC) from the floor of the mouth. The upper panel compares different staining intensities (negative, weak, intermediate, and strong) within each diagnostic group and between different diagnostic groups using Kruskal-Wallis with post hoc Dunn’s tests. Lower panel compares the mean DAB OD, representing sCTLA-4 between the nucleus and the cytoplasm in each of the diagnostic groups in all the cells, *t*-test. The number of images analysed for the top panel per group: normal = 10, LGD = 28, HGD = 13, and SCC = 12. Number of cells analysed in the lower panel per group: normal = 32,197 cells from five images, LGD = 40,510 from four images, HGD = 54,683 from five images, and SCC = 186,131 cells from six images. Error bars represent the standard error of the mean. ** p* < 0.05, ** *p* < 0.01, **** p* < 0.001.

**Figure 3 cancers-15-01696-f003:**
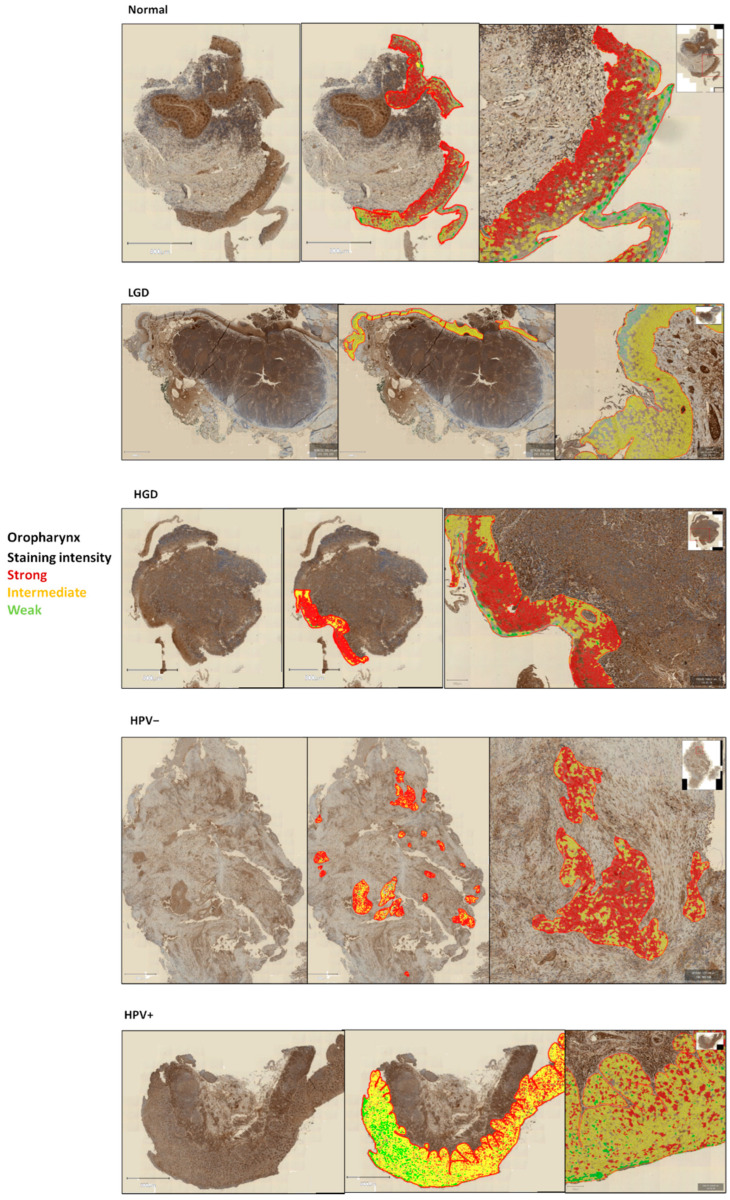
Representative sections from whole-slide scans (X20) of normal, low-grade dysplasia (LGD), high-grade dysplasia (HGD), and HPV− and HPV+ squamous cells from the oropharynx. The left panel shows the immunohistochemical staining for sCTLA-4 (DAB, brown) counterstained with haematoxylin (blue), scale bar represents 800 µm. The middle panel shows the annotations and cell detections with colour-coded staining intensities for the same sections. The right panel represents zoomed-in views of the cell detection, with the scale bar representing 100 µm and the inset showing the exact site of the magnified view.

**Figure 4 cancers-15-01696-f004:**
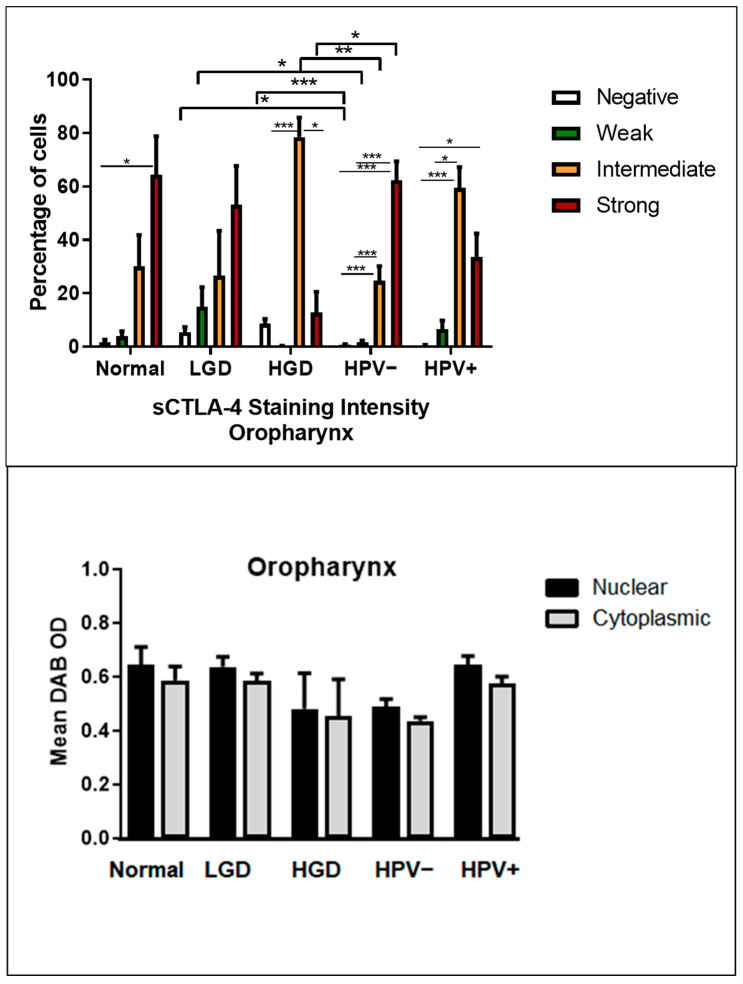
Quantification of the staining intensity in normal, low-grade dysplasia (LGD), high-grade dysplasia (HGD), and HPV− and HPV+ squamous cell carcinoma from the oropharynx. The upper panel compares different staining intensities (negative, weak, intermediate, and strong) within each diagnostic group and between different diagnostic groups using Kruskal-Wallis with post hoc Dunn’s tests. Lower panel compares the mean DAB OD, representing sCTLA-4 between the nucleus and the cytoplasm in each of the diagnostic groups in all the cells. Number of images analysed for the top and middle panels per group: normal = 4, LGD = 5, HGD = 8, HPV-negative SCC = 25, and HPV-positive SCC = 8. Number of cells analysed in the lower panel per group: normal = 36,776 cells from five images, LGD = 33,242 from three images, HGD = 39,515 from two images, HPV− = 210,150 cells from six images, and HPV+ = 136,423 from eight images. Error bars represent the standard error of the mean. ** p* < 0.05, ** *p* < 0.01, *** *p* < 0.001.

**Figure 5 cancers-15-01696-f005:**
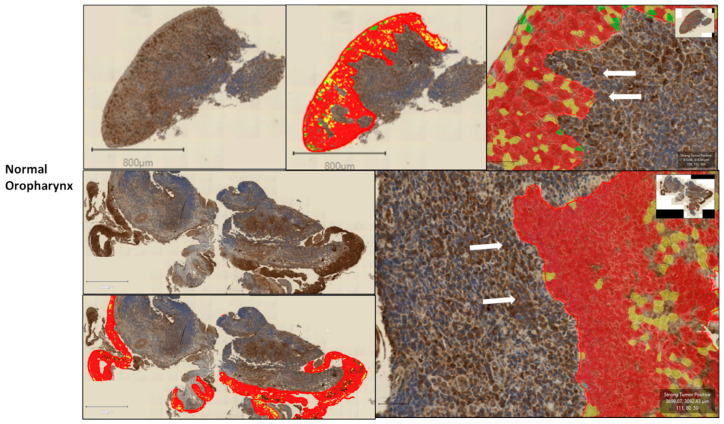
Two examples of normal oropharyngeal epithelia with an underlying heavy inflammatory infiltrate and a loss of the sCTLA-4 staining pattern associated with normal stratification. White arrows point to a subepithelial inflammatory infiltrate. Whole-slide scans (X20); scale bars represent 800 µm in the whole-section images and 50 µm in the zoomed-in image. The top panel is a different scene from the same sample in Appendix A.

**Table 1 cancers-15-01696-t001:** Comparison of staining intensity and H-score between HPV-positive and HPV-negative oropharyngeal SCC.

Parameter	HPV-Negative	HPV-Positive	*p* Value
Negatively stained cells %	0.6257 ± 1.64	0.433 ± 0.6647	0.2004 ^†^
Weakly stained cells %	1.61 ± 3.774	6.538 ± 8.537	***0.011* ^†^**
Intermediately stained cells %	24.74 ± 28.44	59.45 ± 20.58	***0.0087* ^†^**
Strongly stained cells %	62.31± 37.52	33.58 ± 23.27	0.0663 ^†^
H-Score	266.6 ± 37.22	226.1 ± 26.94	***0.0098* ^†^**
Nuclear OD	0.4907 ± 0.0655	0.6454 ± 0.08747	***0.0080* * **
Cytoplasmic OD	0.4359 ± 0.03621	0.5744 ± 0.07428	***0.0007* * **
Nuclear OD (weak)	0.2701 ± 0.01470	0.4117 ± 0.08067	***0.0007* ^†^**
Cytoplasmic OD (weak)	0.2630 ± 0.007861	0.3540 ± 0.02811	***≤0.0001* * **
Nuclear OD (intermediate)	0.4420 ± 0.01041	0.6017 ± 0.05388	***≤0.0001* * **
Cytoplasmic OD (intermediate)	0.3959 ± 0.03170	0.5423 ± 0.03171	***≤0.0001* * **
Nuclear OD (strong)	0.6919 ± 02045	0.8549 ± 0.05206	***0.0426* ^†^**
Cytoplasmic OD (strong)	0.5300 ± 0.02959	0.7441 ± 0.041111	***≤0.0001* * **

^†^ Mann-Whitney test, * Unpaired *t* test. Values in ***bold italics*** indicate statistically significant differences.

## Data Availability

Raw images and electronic whole slide scans can be made available upon request and subject to adherence to ethical approvals.

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
