# Peer review of "A Novel Role for the Soluble Isoform of CTLA-4 in Normal, Dysplastic and Neoplastic Oral and Oropharyngeal Epithelia"

_cancers, 2023, doi:10.3390/cancers15061696_

Round 1

Reviewer 1 Report (Previous Reviewer 2)

The authors describe the role of sCTLA-4 in normal, dysplastic, and neoplastic oral and pharyngeal epithelia. sCTLA-4 staining in the above-mentioned tissues were quantified and distinctive staining patterns observed and described. The authors could show the role of sCTLA-4 in keratozyte differentiation and suggest sCTLA-4 as biomarker in the progression from normal to dysplastic and finally squamous cell carcinoma. The study is well conducted and seems valid and is well written.

The authors have addressed all the relevant points raised by the reviewers. I suggest the editor to accept this study

Reviewer 2 Report (Previous Reviewer 1)

Thank you for addressing my queries. The data regarding the specificity of the sCTLA-4-specific antibody is indeed convincing.

This manuscript is a resubmission of an earlier submission. The following is a list of the peer review reports and author responses from that submission.

Round 1

Reviewer 1 Report

This paper aims to explore the role of sCTLA-4, an alternatively-spliced, soluble, secreted isoform of CTLA-4 (cytotoxic T-lymphocyte associated antigen-4) in head and neck cancer. CTLA-4, and sCTLA-4, have both been shown to have an immunosuppressive function in tumorigenesis by preventing T-cell activation. CTLA-4 is a target for immunotherapy, but antibodies for CTLA-4 also bind the sCTLA-4 isoform. The authors have used a sCTLA-4-specific antibody to explore sCTLA-4 expression in oral and oropharyngeal tissues ranging from normal through dysplastic and malignant samples.

The rationale for the study is very clearly laid out and the methods for staining, analysing and quantifying the immuno-staining data are described in detail and seem appropriate. My main concern is around the sCTLA-4-specific antibody. Has this antibody been validated and, if so, where can this information be found? Since the manuscript is centred on the sCTLA-4-specific antibody for staining tissue samples it is imperative that it is confirmed this antibody is detecting the correct protein. Does this antibody detected a correct sized protein product on western blot and has it been validated by siRNA knockdown? It also appears that two different sCTLA-4 antibodies have been used here. The introduction states that the 73-B1 antibody was used, but according to the methods section this antibody was used for staining the cell lines whilst another antibody – JMW-3B3 – that was developed in-house, was used for staining the tissue sections. Both antibodies should be validated, but especially the in-house one.

It's not clear what staining of the cell lines adds to the manuscript? Particularly as a different antibody was used. Was this to look for differences in localization/expression levels in these SCC cell lines? Shouldn’t there be staining performed on healthy, control cells for comparison? Perhaps analysis of sCTLA-4 expression in cells would be more quantitative via western blot? In combination with cell fractionation the differences between cytoplasmic vs nuclear localization could be explored.

Why is there DAPI staining at the cell membrane in supplementary figures 1 and 2?

For the quantification there are a lot of graphs to look at which makes it difficult to interpret the data. Instead of multiple individual graphs, perhaps it might be easier to display the data in proportion bar graphs, with bars coloured/shaded by the proportion of each staining intensity for each diagnostic group?

The absence of sCTLA-4 in lymphocyte nuclei indicated in figure 5 is hard to discern in these images and it is not made clear why this is an interesting observation?

Minor point:

Is the significance indicator on supplementary figure 4A graph 2 meant to be on graph 3? Graph 2 doesn’t look significant, whilst graph 3 looks like it might be.

Author Response

Comment: This paper aims to explore the role of sCTLA-4, an alternatively-spliced, soluble, secreted isoform of CTLA-4 (cytotoxic T-lymphocyte associated antigen-4) in head and neck cancer. CTLA-4, and sCTLA-4, have both been shown to have an immunosuppressive function in tumorigenesis by preventing T-cell activation. CTLA-4 is a target for immunotherapy, but antibodies for CTLA-4 also bind the sCTLA-4 isoform. The authors have used a sCTLA-4-specific antibody to explore sCTLA-4 expression in oral and oropharyngeal tissues ranging from normal through dysplastic and malignant samples. The rationale for the study is very clearly laid out and the methods for staining, analysing and quantifying the immuno-staining data are described in detail and seem appropriate.

Response:

We would like to thank the reviewer for this summary of the manuscript.

Comment: My main concern is around the sCTLA-4-specific antibody. Has this antibody been validated and, if so, where can this information be found?

Response:

The JMW-3B3 antibody used for IHC has been fully characterized and its specificity to sCTLA-4 reported in Ward FJ, Dahal LN, Wijesekera SK et al. The soluble isoform of CTLA-4 as a regulator of T-cell responses. Eur J Immunol 2013; 43 (5): 1274-1285.

The 73-B1 antibody, which has been raised against the same C terminal linear antigen as JMW-3B3, was the kind gift of Dr David Matthews (MRCT) as part of a collaborative project. We have had the antibody independently tested by a third-party contract research organisation both for specificity against sCTLA-4 and confirmation that it is incapable of binding the receptor isoform of CTLA-4 (unpublished data). For review only, we are showing the data in the attached PDF file

Figure 1. HEK-293 transfected with full length CTLA-4 before analysis of antibody binding to transmembrane CTLA-4 by flow cytometry. As predicted, pan-specific anti-CTLA-4 antibodies bind transmembrane CTLA-4 receptor, whereas anti-sCTLA-4 antibody 73-B1 does not.

We have clarified this in the revised manuscript under the newly added section “Anti-sCTLA-4 antibodies”.

Comment: Since the manuscript is centred on the sCTLA-4-specific antibody for staining tissue samples it is imperative that it is confirmed this antibody is detecting the correct protein. Does this antibody detected a correct sized protein product on western blot and has it been validated by siRNA knockdown?

Response:

We agree with the reviewer that it is important to show that the antibodies bind their target protein.

We can confirm that both JMW-3B3 and 73-B1 recognise affinity purified natural and recombinant sCTLA-4 by Western blot, as well as the data presented above. We were unable to detect any binding in affinity purified flow through.

The JMW-3B3 antibody was fully characterized in Ward FJ, Dahal LN, Wijesekera SK et al. The soluble isoform of CTLA-4 as a regulator of T-cell responses. Eur J Immunol 2013; 43 (5): 1274-1285.

Based on the reviewer’s feedback, we have conducted some experiments using siRNA targeting CTLA-4 (both isoforms) in A549 cells, and our data (now included in supplemental figure1), confirm that 73-B1 is indeed binding the intended target.

We have clarified this in the revised manuscript.

Comment: It also appears that two different sCTLA-4 antibodies have been used here. The introduction states that the 73-B1 antibody was used, but according to the methods section this antibody was used for staining the cell lines whilst another antibody – JMW-3B3 – that was developed in-house, was used for staining the tissue sections. Both antibodies should be validated, but especially the in-house one.

Response:

The two antibodies are interchangeable, as both have been raised against the same C terminal sCTLA-4 specific antigen with the difference being that 73-B1 has a higher affinity than JMW-3B3 for human sCTLA-4.   This has been clarified in the revised manuscript.

Comment:

It's not clear what staining of the cell lines adds to the manuscript? Particularly as a different antibody was used. Was this to look for differences in localization/expression levels in these SCC cell lines? Shouldn’t there be staining performed on healthy, control cells for comparison? Perhaps analysis of sCTLA-4 expression in cells would be more quantitative via western blot? In combination with cell fractionation the differences between cytoplasmic vs nuclear localization could be explored.

Response:

The purpose of staining the three head and neck cancer cell lines was to confirm that head and neck cancers express sCTLA-4, as not all cancer cell lines express this protein.

Upon confirming that it is expressed in cells lines, and once we observed the difference in cytoplasmic and nuclear expression in patient samples in the IHC stained histological sections, we wanted to confirm similar localisation of sCTLA-4 in cell lines and patient samples.

We used confocal microscopy to quantify the localisation of sCTLA-4, but we agree that cell fractionation between cytoplasmic versus nuclear localisation would be informative to explore. Indeed, this is something we are working on in our laboratory across different cell lines and not confined to HNC, but that is beyond the scope of this manuscript.

Comment: Why is there DAPI staining at the cell membrane in supplementary figures 1 and 2?

Response:

The cells were fixed and permeabilised prior to staining with DAPI, and therefore, any A-T rich DNA fragments were stained with DAPI. Furthermore, as we wanted a high confluency of the cells, there would have been some cell debris which would have stained with DAPI as well.

That being said, we refer the reviewer to Supplemental figure 2, where we show that only the nuclei were used for the analysis and any debris were cleaned out of the image prior to the image analysis and quantification of sCTLA-4.

Comment: For the quantification there are a lot of graphs to look at which makes it difficult to interpret the data. Instead of multiple individual graphs, perhaps it might be easier to display the data in proportion bar graphs, with bars coloured/shaded by the proportion of each staining intensity for each diagnostic group?

Response:

We have changed the graphs in figures 2 and 4 and supplemental figure 5 and compiled the multiple panes into one graph. To avoid any loss of information, we opted for the vertical interleaved bar graphs instead of the stacked graph to show the comparisons between the different diagnosis and the different staining intensities within each diagnostic group.

Comment: The absence of sCTLA-4 in lymphocyte nuclei indicated in figure 5 is hard to discern in these images and it is not made clear why this is an interesting observation?

Response:

We agree with the reviewer, and we have now removed this statement. Independent of this manuscript, we are looking at the localisation of sCTLA-4 in immune cells, and this statement does not fall within the remit of the current manuscript.

Comment: Minor point: Is the significance indicator on supplementary figure 4A graph 2 meant to be on graph 3? Graph 2 doesn’t look significant, whilst graph 3 looks like it might be.

Response:

We have checked the data and confirmed that the statistical significance indicators in the figure are accurate. There was no difference below p<0.05 in graph 3 (the weak staining), and in graph 2 (negative staining), there was a significant difference between Dysplasia and SCC (p<0.05).

Reviewer 2 Report

The authors describe the role of sCTLA-4 in normal, dysplastic, and neoplastic oral and pharyngeal epithelia. sCTLA-4 staining in the above-mentioned tissues were quantified and distinctive staining patterns observed and described. The authors could show the role of sCTLA-4 in keratozyte differentiation and suggest sCTLA-4 as biomarker in the progression from normal to dysplastic and finally squamous cell carcinoma. 

The study is well conducted and seems valid and is well written. I suggest the editor to accept this study. 

Author Response

We would like to thank the reviewer for the positive feedback.